# A CircRNA–miRNA–mRNA Network for Exploring Doxorubicin- and Myocet-Induced Cardiotoxicity in a Translational Porcine Model

**DOI:** 10.3390/biom13121711

**Published:** 2023-11-27

**Authors:** Julia Mester-Tonczar, Patrick Einzinger, Ena Hasimbegovic, Nina Kastner, Victor Schweiger, Andreas Spannbauer, Emilie Han, Katrin Müller-Zlabinger, Denise Traxler-Weidenauer, Jutta Bergler-Klein, Mariann Gyöngyösi, Dominika Lukovic

**Affiliations:** 1Division of Cardiology, Department of Internal Medicine II, Medical University of Vienna, 1090 Vienna, Austria; julia.mester-tonczar@meduniwien.ac.at (J.M.-T.); ena.hasimbegovic@meduniwien.ac.at (E.H.); nina.kastner@meduniwien.ac.at (N.K.); victor.schweiger@gmail.com (V.S.); andreas.spannbauer@meduniwien.ac.at (A.S.); katrin.zlabinger@meduniwien.ac.at (K.M.-Z.); denise.traxler-weidenauer@meduniwien.ac.at (D.T.-W.); jutta.bergler-klein@meduniwien.ac.at (J.B.-K.); mariann.gyongyosi@meduniwien.ac.at (M.G.); 2Research Unit of Information and Software, Institute of Information Systems Engineering, 1040 Vienna, Austria; patrick.einzinger@aon.at

**Keywords:** circRNA, miRNA, target prediction, doxorubicin, Myocet, network analysis, porcine model

## Abstract

Despite the widespread use of doxorubicin (DOX) as a chemotherapeutic agent, its severe cumulative cardiotoxicity represents a significant limitation. While the liposomal encapsulation of doxorubicin (Myocet, MYO) reduces cardiotoxicity, it is crucial to understand the molecular background of doxorubicin-induced cardiotoxicity. Here, we examined circular RNA expression in a translational model of pigs treated with either DOX or MYO and its potential impact on the global gene expression pattern in the myocardium. This study furthers our knowledge about the regulatory network of circRNA/miRNA/mRNA and its interaction with chemotherapeutics. Domestic pigs were treated with three cycles of anthracycline drugs (DOX, *n* = 5; MYO, *n* = 5) to induce cardiotoxicity. Untreated animals served as controls (control, *n* = 3). We applied a bulk mRNA-seq approach and the CIRIquant algorithm to identify circRNAs. The most differentially regulated circRNAs were validated under cell culture conditions, following forecasting of the circRNA–miRNA–mRNA network. We identified eight novel significantly regulated circRNAs from exonic and mitochondrial regions in the porcine myocardium. The forecasted circRNA–miRNA–mRNA network suggested candidate circRNAs that sponge miR-17, miR-15b, miR-130b, the let-7 family, and miR125, together with their mRNA targets. The identified circRNA–miRNA–mRNA network provides an updated, coherent view of the mechanisms involved in anthracycline-induced cardiotoxicity.

## 1. Introduction

Anthracyclines are essential in cancer therapy, and while they improve the prognosis of various cancer types, their clinical use is limited due to dose-dependent cardiovascular toxicity [1]. As a result, patients undergoing anthracycline-based therapies may experience doxorubicin (DOX)-induced side effects, which may include chronic, life-threatening cardiomyopathy or other cardiac complications [2].

To address this issue, the liposomal encapsulation of DOX (Myocet, MYO) has emerged as a valuable alternative, reducing cardiotoxicity by minimizing myocardial drug accumulation [3]. Numerous studies have described the potential molecular mechanisms of cardiotoxicity, including Ca^2+^ handling abnormalities, the inhibition of topoisomerase IIβ, the generation of reactive oxygen species (ROS), the boosting of interferon-related DNA damage resistance, and mitochondrial dysfunction. Particularly oxidative stress arising from the mitochondria has been described as a DOX-dependent cardiotoxic mechanism [4,5,6].

Recently, research has increasingly focused on the role of circulating non-coding RNAs since the expression of circular RNAs (circRNAs) [7], long non-coding RNAs, and microRNAs (miRNAs) subsequently affects the regulation of molecular pathways [8,9,10]. CircRNAs play pivotal roles in cancer development by either inhibiting or promoting the growth and metastasis of various cancer types [11,12]. Their circular structure and sequence homology promote covalent binding with circulating proteins and miRNAs, which they may sponge. Furthermore, these circular molecules also represent promising biomarkers for various cancers, particularly breast [13], colorectal [14,15], or gastric cancer [15,16]. However, little is known about the effect of chemotherapeutics on circRNAs’ expression.

The capacity of circRNAs to interact with miRNAs suggests their potential as a novel approach in therapy for anthracycline-induced cardiotoxicity or as a biomarker for preventive strategies. CircITCH sponging miR-330-5p [8] or the circPan3–miR-31-5p axis [16] has revealed its potential to regulate doxorubicine-induced cardiotoxicity.

A recent study has demonstrated that circular RNA can have a translational impact on the prevention of doxorubicin-induced cardiotoxicity. Circular RNA was derived from the insulin-receptor (INSR)-mediated cardioprotection in doxorubicin-treated human and rat cardiomyocytes [17]. The application of advanced sequencing techniques and bioinformatic tools enables the identification of novel circRNAs and the study of their interactions in order to offer a complex view of challenging therapeutics. Additionally, novel prediction algorithms and bioinformatics pipelines facilitate the construction of networks of interacting molecules, leading to the identification of key molecules associated with critical phenotypes [18,19].

In this study, we investigated the commonly used chemotherapeutics DOX and MYO, and we examined how they affect circRNA expression in vivo in a pig model and in vitro in porcine cardiac progenitor cells (pCPCs) and porcine cardiac fibroblasts (pCFs). Our results demonstrate the significant downregulation of two mitochondrial-genome-derived circRNAs (mt-circRNAs) in vivo in pigs treated with DOX and, to a lesser extent, with MYO compared to the control group (CO) (*p* < 0.001), as confirmed via qPCR analysis. Additionally, we identified novel genome-derived circRNAs and constructed a sequence-based network of interacting circRNA–miRNA–mRNA using bioinformatics prediction tools and databases, which were further validated with mRNA-seq bulk sequencing data retrieved from the same animal models. These newly described mt-circRNAs and circRNAs offer new insights into the mechanisms of chemotherapy-induced cardiotoxicity.

## 2. Materials and Methods

### 2.1. Ethics Approval

This animal study was conducted at the Institute of Diagnostic Imaging and Radiation Oncology of the University of Kaposvar (Kaposvar, Hungary) and was approved by the ethics committee (EK:1/2012-KEATK DOI, MUW §27 Project FA714B0518). The animal handling procedures were in accordance with the *Guide for the Care and Use of Laboratory Animals* [20].

### 2.2. Animal Study

The detailed experimental protocols have previously been published [21]. Briefly, healthy domestic pigs (*Sus scrofa domesticus*) (3 months old with weights ranging from 25 to 30 kg) were randomized into treatment groups that received intravenous infusions of either DOX (*n* = 6), MYO (*n* = 5) in doses equivalent to human medication regimens (60 mg/m^2^ body surface area), or a vehicle (control, *n* = 5). For the intravenous treatment, the animals were sedated with 12 mg/kg of ketamine hydrochloride, 1 mg/kg of xylazine, and 0.04 mg/kg of atropine. Due to the deteriorating health of the pigs, the experiment was prematurely terminated. Premature death occurred in one DOX pig, resulting from leucopenia and thrombocytopenia, and one MYO pig with renal failure. The deceased animals in the DOX and MYO groups had elevated levels of NT-proBNP and Troponin I. Hemorrhagic pericarditis in the right ventricle of one DOX pig was revealed after obduction.

In this study, we conducted a secondary analysis of circular RNA (circRNA) expression using left ventricular (LV) tissue samples obtained from the work previously published by Gyöngyösi et al. [21]. Cardiotoxicity assessments in the original study were performed through a combination of methods, including the measurement of established cardiotoxicity biomarkers, histological staining, and the evaluation of cardiac function. Notably, all animals treated with DOX and MYO in our study displayed significantly elevated levels of creatinine kinase, a well-recognized marker of cardiac injury. Moreover, manifestations of cardiotoxicity were evident in the histological analysis, characterized by the disorientation of the heart muscle fibers. Additionally, both left ventricular (LV) and right ventricular (RV) functions exhibited marked decreases, further underscoring the presence of cardiotoxic effects induced via the DOX and MYO treatments.

The animals were euthanized under continuous anesthesia (1.5–2.5 vol.% isoflurane, 1.6–1.8 vol.% O_2_, and 0.5 vol.% N_2_O) using an intravenous dose of heparin (10,000 U) and 10 mL of intravenous saturated KCl (10%). The hearts were explanted, and tissue samples of the left ventricles (LVs) (150–250 mg) were collected and stored in RNAlater (Thermo Fisher Scientific, Waltham, MA, USA) at −80 °C.

Since, in both treatment groups (DOX and MYO), premature death occurred, in this study, we used a sample size of *n* = 5 in the DOX and MYO groups. For our control group, we used *n* = 3 in this study because, in our previous experiment, tissue from 2 control animals had already been consumed. Tissue from the LVs of the DOX and MYO groups was used for circRNA sequencing. Additionally, we validated the data through qPCR, including LV tissue from the control, DOX, and MYO groups.

### 2.3. RNA Extraction from Tissue

The thawed LV tissue samples were cut (25–30 mg) and placed into Precellys CK28 tubes containing the Qiazol Lysis Buffer (Qiagen, Hilden, Germany). The tissue was homogenized using the program 5000:3 × 20 − 20 s on the Precellys 24 (Bertin, Rockville, MD, USA). The supernatant was then transferred into 2-mL tubes containing 200 µL of chloroform. The tubes were then centrifuged at 11,600× *g* at 4 °C for 15 min, and the upper aqueous phase was then transferred to new 2-mL tubes. Total RNA was isolated using the miRNeasy Mini kit (Qiagen), including DNase I digestion, following the manufacturer’s protocol. Total RNA quality and quantity were assessed using the Nanodrop One (Thermo Fisher Scientific, Karlsruhe, Germany) and the 2100 Bioanalyzer (Agilent, Vienna, Austria).

### 2.4. RNA Sequencing

Isolated total RNA was sequenced using the Illumina NextSeq2000 sequencing approach. The circRNA sequencing samples had a mean depth of 90 million paired-end (150 bp) reads. In order to accurately sequence circRNAs, no poly(A) enrichment was performed. Further, mRNA enrichment was achieved using the NEBNext rRNA depletion kit (New England Biolabs, Ipswich, MA, USA).

Bulk mRNA sequencing was previously conducted, and it aimed to investigate mechanisms involved in DOX- and MYO-induced cardiotoxicity [21]. In the current study, our analysis centered on the comparison between DOX and MYO, utilizing the bulk mRNA-sequencing data obtained from Gyöngöysi et al. as a reference for the validation of predicted gene targets [21]. The NEBNext Poly(A) mRNA Magnetic Isolation Module from New England Biolabs was employed. Following mRNA priming and fragmentation, reverse transcription was performed, and a cDNA library was synthesized using the NEBNext Ultra Directional RNA Library Kit, also from New England Biolabs. The HiSeq 2500 platform from Illumina in San Diego, CA, USA, was utilized for the sequencing process, employing paired-end reads with a length of 50 base pairs. Each library underwent sequencing to achieve a mean depth of 17.7 million (standard deviation: 2.9 million) paired-end reads (Appendix A). Library preparation and sequencing were conducted at the VBCF NGS Unit (Vienna Biocenter Core Facilities, www.viennabiocenter.org/facilities (accessed on 1 January 2022)).

### 2.5. Bioinformatics

The raw sequencing reads (FASTQ files) were trimmed with Trim Galore (default values), and quality control was performed using FastQC. We then used the CIRIquant algorithm [22], which allows the quantification and differential analysis of circRNAs in a single pipeline. It incorporates the HISAT2 algorithm for the alignment of reads from RNA-seq, CIRI2 for the identification of circRNAs, and edgeR (v3.32.1) for detecting the significant differential expression of circular transcripts. We used the reference genomes and annotation files from iGenomes (Ensembl, Sus scrofa 11.1 .gtf file). The CIRIquant algorithm requires an annotated fasta reference genome with fai indices, a gtf annotation file, bwa, and hisat indices. The quantification of circRNA requires a list of junction sites in the bed format. Similarly, a bulk mRNA-seq pipeline was initiated after demultiplexing, trimming, FastQC, and alignment to the Sus scrofa 11.1 (Ensembl, .gff file) genome. The EdgeR algorithm was employed to identify statistically significant, differentially regulated mRNA targets within the bulk mRNA-seq dataset (v3.32.1).

### 2.6. Network Analysis

A regulatory network involving circRNA–miRNA–mRNA interactions was constructed using Cytoscape software (version 3.9.1). CircRNAs with validated expression through qPCR and cell culture experiments were employed as a primary input for the network analysis. We selected seven significantly regulated circRNAs (*p*.adj.value < 0.05) with both upregulated (LogFC > 6) and downregulated expression (LogFC < −7) in the comparison between the DOX and MYO groups (Appendix A). To identify circRNA-associated microRNAs, the custom prediction tool MiRanda (v3.3a) was employed [23]. The following criteria were applied for prediction: an open penalty (−9.0), a gap extend penalty (−4.0), a score threshold (140), an energy threshold (−25.0), and a scaling parameter (4.0). In order to identify clinically relevant miRNA, human orthologs of the predicted miRNA molecules were obtained from the HGNC database (https://www.genenames.org/tools/hcop/ (accessed on 1 March 2023)).

The obtained miRNAs were analyzed using the miRwalk database (MiRWalk 2.0) to predict the target mRNA genes [24]. mRNA targets with a score exceeding 0.8 and exhibiting overlapping matches in the TargetScan, MiRDB, and miRTarBase databases were subsequently included in the network analysis.

Finally, the predicted data were filtered with bulk mRNA-seq data retrieved from the LV tissue of the identical DOX- and MYO-treated animals. The resulting network comprised circRNA–miRNA and mRNA genes that matched with bulk mRNA-sequencing results (FDR < 0.05). The MiRanda prediction algorithm identified some redundant microRNAs that were present in both settings, involving downregulated and upregulated circRNAs. To create a circRNA–miRNA–mRNA network with unique molecules (microRNA and mRNA), we merged the results and represented both downregulated and upregulated circRNAs in a single network.

A web server g:Profiler was utilized to perform a gene ontology enrichment analysis of overlapped genes with parameters of *p* < 0.05 and counts > 2 [25]. The bulk mRNA-sequencing data discussed in this publication were deposited in NCBI’s Gene Expression Omnibus and are accessible through GEO Series accession number GSE197049.

### 2.7. Primary Cell Lineages

For our validation experiments, we used porcine cardiac progenitor cells (pCPCs) and porcine cardiac fibroblasts (pCFs). The pCPCs were isolated as described in our previous publication [26]. The pCFs were purchased from Cell Biologics (Chicago, IL, USA). For our in vitro experiments, we used *n* = 6 for technical replicates.

### 2.8. In Vitro Treatment of Doxorubicin and Myocet

We have previously published the dose-finding study for DOX [21]. The cells were then treated with DOX (1.56 nM (*n* = 6) and 6.25 nM (*n* = 6)) and MYO (1 µM (*n* = 6) and 0.1 µM (*n* = 6)) for 24 h. Non-treated pCPCs and pCFs (both *n* = 6) served as controls. After 24 h, cytotoxicity was assessed (EZ4U Assay, Alpco Diagnostics, Salem, NH, USA) using the cells in a 96-well plate. The cells that were seeded in 6-well plates were stored in an RLT buffer (Qiagen) with beta-mercaptoethanol (1:100) at −80 °C until RNA isolation.

### 2.9. RNA Isolation from Cells

Cells in the RLT buffer were transferred into QIAshredder tubes (Qiagen) and centrifuged for 2 min at 16,000× *g* and 4 °C. The flow-through was subjected to automated RNA isolation (QIAcube, Qiagen) using the miRNeasy Mini kit (Qiagen).

### 2.10. cDNA Synthesis and Real-Time Quantitative PCR Analysis

cDNA synthesis was performed using the QuantiTect Reverse Transcription kit (Qiagen) according to the manufacturer’s protocol. qPCR was performed using the QuantiTect SYBR Green PCR kit (Qiagen) on the QuantStudio 5 Real-Time PCR System (Thermo Fisher Scientific) with two technical repetitions. The primers spanning the backsplice junction are listed in Appendix A. For tissue sample analysis, circRNA expression levels and the expression of Cas3 and Ki67 were normalized to the geometric mean of ACTB and HPRT. For our in vitro experiments (both pCPCs and pCFs), circRNA expression levels were normalized to HPRT.

### 2.11. Statistics

For the qPCR results of cardiac tissue and cell lines (CF and pCPC), we performed differential expression analysis via a one-way analysis of variance (ANOVA) of log2 fold changes, which were calculated with the ∆∆Ct method. For each combination (circRNA, cell type/tissue, and group), the data were checked for normality with the Shapiro–Wilk test. Furthermore, we tested for heteroscedasticity with Levene’s test. We performed a standard ANOVA (with pairwise *t*-tests for post hoc analysis) if its assumptions were met or the Kruskal–Wallis test (with a pairwise Wilcoxon signed-rank test for post hoc analysis) otherwise. The Benjamini–Hochberg method was used to correct for multiple testing. Two-sided *p*-values of <0.05 were considered significant.

## 3. Results

### 3.1. Identification of circRNAs in Pig Hearts Using CIRIquant

Using the CIRIquant algorithm, we identified differentially expressed cardiac circRNAs in pigs treated with either DOX or MYO, with untreated pigs serving as controls. Most of the identified circRNAs were exonic, but the most significant ones were derived from the mitochondrial genome, as revealed via CIRIquant.

From our RNA-seq data, we selected ten circRNAs that exhibited significant dysregulation between the DOX–MYO groups (Appendix A). A subsequent qPCR analysis of porcine heart tissue confirmed significant downregulation of circ-MT:3033|3289 in the DOX control group (*p* < 0.001) and MYO control group (*p* < 0.001) (Figure 1A). Similarly, circ-MT:3070|3478 showed downregulation in both the DOX control (*p* < 0.001) and MYO control (*p* < 0.001) groups (Figure 1B). Additionally, circ-16 displayed downregulation in the DOX control (*p* = 0.045) and MYO–DOX (*p* = 0.048) groups (Figure 1C).

### 3.2. Concentration Finding of Doxorubicin and Myocet in pCPCs and pCFs In Vitro

To determine the appropriate MYO concentration for in vitro treatment, we tested different concentrations of MYO ranging from 0.1 µM to 5 µM in pCPCs. This concentration range was based on the findings of Toldo et al., which indicated that it was well tolerated both in vivo and in vitro [27]. Our cytotoxicity assay revealed minimal to moderate cytotoxicity in pCPCs treated with 0.1 µM and 1 µM of MYO, while 3 µM and 5 µM showed a high level of cytotoxicity in pCPCs (Appendix A).

We also examined the cytotoxicity of DOX and MYO in pCPCs (Appendix A) and pCFs (Appendix A). The DOX concentrations used in our cytotoxicity assay for pCPCs and pCFs were based on these results (Appendix A). A four-fold increase in the concentration of DOX in the cell culture experiments led to a three-fold increase in the proportion of cells exhibiting cytotoxic activity.

Interestingly, even though our DOX doses (6.25 nM and 1.56 nM) were lower than our MYO doses (1 µM and 0.1 µM), we observed similar levels of cytotoxicity, leading us to conclude that DOX exhibits higher cytotoxic effects in both primary cell lineages compared to MYO-treated cells (Appendix A).

### 3.3. CircRNAs in DOX and MYO-Treated pCPCs and pCFs

We examined the expression of two circRNAs that showed significant dysregulation between MYO and DOX treatments compared to the controls. In pCPCs, the expression of circ-MT:3033|3289 was downregulated in MYO-treated cells compared to the control and the DOX treatment (*p* < 0.001 for MYO 1 µM compared to the control, *p* = 0.003 for MYO 1 µM compared to DOX 1.56 nM, and *p* < 0.001 for MYO 0.1 µM compared to MYO 1 µM) (Figure 2A). Conversely, in pCFs, the expression of circ-MT:3033|3289 was significantly upregulated in the MYO and DOX treatment groups compared to the control (*p* = 0.0435 for MYO 1 µM compared to the control, *p* = 0.002 for MYO 1 µM compared to DOX 1.56 nM, *p* = 0.0032 for MYO 1 µM compared to DOX 6.25 nM, *p* = 0.0037 for MYO 0.1 µM compared to DOX 6.25 nM, and *p* = 0.002 for MYO 0.1 µM compared to DOX 1.56 nM) (Figure 2B). However, no significant changes were observed in the expression of circ-MT:3070|3478 in either pCPCs or pCFs (Figure 2C,D). These results indicate that MYO treatment influences the expression of circ-MT:3033|3289 more significantly than DOX treatment in the tested cell types. The backsplice junctions of mitochondrial circRNAs were confirmed via Sanger sequencing (Appendix A).

### 3.4. Expression of Cas3 and Ki67 in DOX- and MYO-Treated pCPCs and pCFs

In order to assess whether the administration of DOX and MYO triggers apoptosis, an event relevant to doxorubicin-induced cardiotoxicity, we analyzed the expression of the apoptotic markers Cas3 and Ki67 in pCPCs and pCFs. In pCPCs, Cas3 was significantly upregulated in cells treated with 0.1 µM of MYO compared to the control (*p* = 0.022) and compared to DOX 6.25 nM (*p* = 0.022) (Figure 3A). However, no significant differences in Cas3 expression were observed in the DOX- or MYO-treated pCFs (Figure 3B).

Regarding the proliferation marker Ki67, in pCPCs, Ki67 was significantly upregulated in the MYO treatment groups compared to the control (*p* = 0.007 for MYO 0.1 µM and *p* = 0.017 for MYO 1 µM). Moreover, in the 1 µM MYO treatment group, Ki67 was significantly upregulated compared to the DOX treatment groups (*p* = 0.017 for DOX 1.56 nM and *p* = 0.025 for DOX 6.25 nM) (Figure 3C). In addition, this difference already appeared at a lower dosage of MYO (0.1 µM) compared to both the DOX groups (Figure 3C,D). Similarly, in pCFs, Ki67 was significantly upregulated in the MYO treatment groups compared to the control group (*p* = 0.049 for MYO 0.1 µM and *p* < 0.001 for MYO 1 µM). Additionally, Ki67 expression in pCFs was significantly upregulated between the MYO and DOX treatment groups (*p* = 0.043 for MYO 0.1 µM compared to DOX 1.56 nM, *p* < 0.001 for MYO 1 µM compared to DOX 1.56 nM, and *p* < 0.001 for MYO 1 µM compared to DOX 6.25 nM) (Figure 3D).

These findings indicate significant differences in the expression of Cas3 between the MYO and DOX treatments in pCPCs, suggesting distinct effects of the two chemotherapeutic agents on apoptosis in this primary cell line (Figure 3A). No significance was detected in the expression of Cas3 between the MYO and DOX treatments in pCFs (Figure 3B). The expression of Ki67 was significantly regulated in both pCPCs and pCFs (Figure 3C,D), leading to the conclusion that MYO and DOX affect cell proliferation in these primary cell lines. However, one has to keep in mind that Myocet and doxorubicin have the same chemical agent, but Myocet contains the chemical agent in an encapsulated form. Therefore, these distinct effects could be due either to the encapsulation, which makes Myocet less toxic, or the fact that different concentrations of DOX and MYO were used in this experiment.

### 3.5. CircRNAs-miRNA-Gene Network Analysis

By employing prediction algorithms for miRNA identification, we constructed a network of interacting microRNAs that exhibited homology in circRNA sequences and their corresponding targets (Appendix A). Only verified human orthologs underwent further prediction for mRNA targets (Appendix A). We identified 126 overlapping genes by intersecting the miRNA targets with differentially expressed genes retrieved via bulk RNA-seq (Appendix A), with which we compared the DOX and MYO groups.

To analyze the regulatory network, we merged two independent networks of significantly upregulated and downregulated circRNAs from the CIRIquant pipeline, resulting in a single circRNA–miRNA–mRNA regulatory network. This allowed us to identify key sponged microRNAs with corresponding mRNA targets from bulk mRNA-seq (Appendix A).

In particular, upregulated circRNA1 (chr7:22854938|22958923), downregulated circRNA6 (chr15:71259536|71309260), and circRNA4 (chr7:22941376|22958923) showed interactions with miR-17 and let7 isoforms, which target predominantly downregulated genes (FDR < 0.05) in the MYO group. Our functional analysis of the validated target genes associated with miR-17 and the let7 family revealed significantly regulated pathways related to cell cycle and transcriptional activity. These pathways are known to be central to the mechanisms of action of anthracyclines (Appendix A). This suggests a significant role of the newly identified circRNAs and their targets in reducing cardiotoxicity resulting from MYO treatment compared to DOX.

Furthermore, utilizing the MiRanda prediction algorithm, we identified miR-130b-5p and miR-9829-5p targeting mt-circRNA 3033|3289, and miR-4339 targeting mt-circRNAs 3070|3478. Among these, only miR-130b-5p exhibited a human orthologous sequence. Subsequently, MiRwalk analysis computationally identified 177 miR-130b-5p-target genes (Figure 4A) that were compared to mRNA-sequencing data (Figure 4B). One hundred and seventy-seven target genes are involved in PI3K-signaling and the regulation of the actin cytoskeleton (Appendix A). Highly interconnected nodes between circRNA–miRNAs and predicted targets, which were also present in the mRNA-sequencing data, indicate the relevance of miRNAs in anthracycline-induced cardiotoxicity.

## 4. Discussion

A growing body of research has highlighted the significant role of circRNAs in doxorubicin-induced cardiotoxicity, in which they act as sponges for miRNAs that target apoptosis [28], pro-inflammatory pathways [9], and cellular/mitochondrial oxidative stress [8]. In our study, we aimed to investigate the regulatory circRNA/miRNA/mRNA network’s involvement in the distinct cardiotoxicity profiles of the Myocet model compared to doxorubicin treatment. We analyzed circRNA expression patterns in an in vivo translational pig model treated with DOX or MYO and in vitro with DOX and MYO-treated pCPCs and pCFs.

The investigated drugs, doxorubicin and its liposomal formulation, Myocet, operate by entering the intracellular milieu via the plasma and nuclear membranes, subsequently intercalating with nucleic acids. This process leads to the inhibition of DNA synthesis, which, in turn, orchestrates apoptosis and cell death [29]. Since doxorubicin and Myocet do not exhibit specificity toward particular cell types and instead induce cell cycle arrest in dividing cells, we utilized porcine cardiac progenitor cells (pCPCs) and porcine cardiac fibroblasts (pCFs) isolated from left ventricular tissues for in vitro cell culture experiments.

Using next-generation sequencing approaches, together with a novel bioinformatic pipeline encompassing the CIRIquant algorithm [22] and prediction tools for target identification, we were able to detect significantly regulated circRNA, together with relevant microRNAs and gene targets that may be involved in anthracycline-induced cardiotoxicity. In addition, the suggested sequences were cross-validated with qPCR, cell culture experiments, and bulk mRNA-seq data. To present data with clinical value, we selected only human orthologs that can be applied in further human studies.

Our in vivo findings demonstrated a significant downregulation of mt-circRNAs in animals treated with DOX and MYO compared to the control group. However, our in vitro experiments revealed a discrepancy between the primary cell culture lines and the data obtained from the in vivo model, particularly regarding the expression of circ-MT:3033|3289. This disparity could be attributed to cell line-specific expression patterns that may not fully reflect the circRNA expression in the myocardium. In the myocardium, various cell types may influence the deregulation of circRNAs, leading to differences between in vitro and in vivo results.

Furthermore, a study conducted by Everaert et al. compared different RNA-seq pipelines to qPCR results. Their research indicated that, while most genes showed similar expression patterns in both RNA-seq data and qPCR results, approximately 15–20% of genes exhibited either differential expression in opposing directions or were detected as differentially expressed by one method but not the other [30,31]. Gene expression data results from a highly complex analysis, and the comparison of different quantification methods requires careful interpretation and consideration.

Our prediction algorithm detected a significant interaction between circ-MT:3033|3289 and miR-130b-5p. The constructed network of circ-MT-RNA/miRNA/mRNA identified common targets, which were subsequently verified using bulk mRNA-seq. Further analysis revealed that these targets are involved in the regulation of the actin cytoskeleton and PI3K signaling pathways. Interestingly, a recent study by Coban et al. found an elevated expression of miR-130b-5p in blood cells from patients with coronary artery disease (CAD), leading them to propose miR-130b-5p as a potential prognostic biomarker for CAD severity [32]. The increased expression of miR-130b was negatively related to DOX sensitivity in patients with bladder carcinoma. [33]

Furthermore, we observed that differences in circRNA expression were only evident at specific doses (Figure 2A,B). Both Figure 2A and Figure 2B clearly show a significant difference between MYO-treated and control cells. Notably, achieving a comparable level of cardiotoxicity to DOX required a much higher concentration of MYO (6.25 nM of DOX vs. 1 µM of MYO and 1.56 nM of DOX vs. 0.1 µM of MYO), once again confirming that MYO is less toxic than DOX (Appendix A).

In recent years, circRNAs have been the subject of numerous studies investigating the pathophysiology of various conditions, ranging from cancer to cardiovascular disease [34,35,36,37]. However, research on non-coding RNAs originating from the mitochondrial genome is relatively new, with initial studies published as recently as 2020 [38] and 2022 [39]. Consequently, limited information is available regarding their physiological function or their roles in the pathophysiology of specific conditions.

Despite extensive research into the molecular mechanisms of DOX-induced cardiotoxicity, the precise mechanisms involved have not been conclusively elucidated. Recent studies have particularly focused on the involvement of mitochondria in doxorubicin-induced cardiotoxicity. One study demonstrated that DOX can trigger mitochondria-dependent ferroptosis, a significant contributor to DOX-induced cardiotoxicity [40]. Another study highlighted the role of mitochondria in mediating doxorubicin-induced cardiotoxicity, where DOX was shown to lead to iron accumulation inside the mitochondria [41].

A 2020 study reported that mt-circRNAs act as molecular chaperones, facilitating protein entry into the mitochondria. Additionally, it found that mt-circRNAs are crucial for the adaption of mitochondria to physiological conditions and diseases by modulating mitochondrial protein importation [42]. These findings shed light on the potential significance of mt-circRNAs in mitochondrial function and their possible implications in various physiological and pathological processes involved in anthracycline-induced cardiotoxicity.

We have successfully identified ten novel circRNAs expressed in the myocardium of DOX- and MYO-treated animals, demonstrating significant upregulation and downregulation. These findings, in conjunction with their predicted miRNA targets, hold promise as potential prognostic biomarkers for doxorubicin-induced cardiotoxicity.

Our miRNA prediction pipeline allowed us to detect a set of miRNAs, which includes miR-17, various isoforms of let-7, miR125, miR-15b, and miR-130b, along with their respective mRNA targets. Furthermore, we were able to validate the presence of these miRNAs in both the MYO and DOX models. Our study aligns with prior research, as downregulated miR-17 and the let-7 family have previously been associated with doxorubicin-induced cardiotoxicity in a study conducted on rat cardiomyocytes. Within a similarly constructed network, it identified these miRNAs as critical nodes targeting genes involved in the regulation of the extracellular matrix and cell survival [43]. This observation suggests their potential role in the underlying mechanisms of anthracycline-induced cardiotoxicity.

Previous studies exploring the mechanistic insights into doxorubicin-induced cardiotoxicity have proposed the involvement of the let7 family [43] and several circRNAs [8,9,28]. In contrast, our study constructed a comprehensive circRNA/miRNA/mRNA regulatory axis based on prediction algorithms. Subsequently, we validated the expression of the circRNAs using a qPCR approach and further confirmed the expression of two mitochondrial circRNAs under cell culture conditions in vitro. The in silico predictions were verified by cross-referencing them with the bulk mRNA-seq data and the predicted target genes. As a result, we were able to provide a detailed and systematic view of the mechanisms implicated in doxorubicin-induced cardiotoxicity. Furthermore, the gene ontology enrichment analysis of selected mRNAs from the network demonstrated a significant involvement of genes associated with DNA-binding transcription activation and the cell cycle. The fact that anthracyclines specifically target the cellular processes responsible for DNA replication and transcription [43] suggests a substantial interrelation between the chosen miRNAs (miRNA-17 and the let-7 family) and the development of anthracycline-induced cardiotoxicity. The utilization of specific miRNAs as prospective biomarkers for anthracycline-mediated myocardial damage, which was identified in the pig model, was justified by our emphasis on human orthologs that are also present in the human transcriptome. The constructed network of non-coding molecules, including circRNA and miRNA, along with their respective gene targets, introduces novel opportunities to utilize sequencing data to identify potential biomarkers that possess predictive value in anthracycline-induced cardiotoxicity [7,18,44]. It is imperative to investigate whether these expressed molecules are also detectable in patients who exhibit increased cardiac susceptibility resulting from early-stage anthracycline treatment.

In our previous study, we elaborated upon the porcine translational animal models of anthracycline-induced cardiotoxicity [21]. The study involved the validation of cardiotoxicity and the measurements of decreased cardiac function using cardiac magnetic resonance imaging (cMRI) and echocardiography. We conducted a comprehensive assessment of the cardiotoxic effects at the tissue level through histology staining, ELISA measurements, and cytotoxicity assays. By employing a bulk mRNA-sequencing approach, we assessed the mechanisms involved in DOX- and MYO-induced cardiotoxicity [21]. The current study represents a secondary analysis of circular RNA expression within the same porcine model. This analysis led us to develop a pipeline that includes prediction algorithms and network analysis, enabling us to search for molecular biomarkers related to this translational model.

## 5. Conclusions

This study has demonstrated that both DOX and MYO treatments significantly influence the expression of circRNAs and mt-circRNAs, both in vivo and in vitro. The constructed circRNA–miRNA–mRNA network revealed the strong involvement of miR-17, miR-15b, miR-130b, and let7-isoforms, along with their gene targets, enhancing our understanding of anthracycline-induced cardiotoxicity. In conclusion, circRNAs derived from the mitochondrial genome provide novel insights into the cardiotoxic effects of chemotherapeutics and represent potential targets for mitigating cardiac damage caused by anthracyclines.

## 6. Limitations

The predicted miRNAs were not experimentally validated using a qPCR approach, and further investigations are required to validate the expression levels of the identified miRNAs in the DOX and MYO models. Given that miRNA expression is subject to regulation via multiple mechanisms, no direct interaction between miRNAs and mRNA targets was evaluated.

As the sequence homology of circRNAs can vary between *Sus scrofa* and *Homo sapiens*, further validation of the cellular expression of particular circRNAs in cell culture experiments involving human cell lines is necessary. However, circRNAs in the network analysis served as a primary data point for the downstream prediction and validation pipeline, and only human orthologs were selected for the construction of the network in order to maintain a translational concept for the study.

## Figures and Tables

**Figure 1 biomolecules-13-01711-f001:**
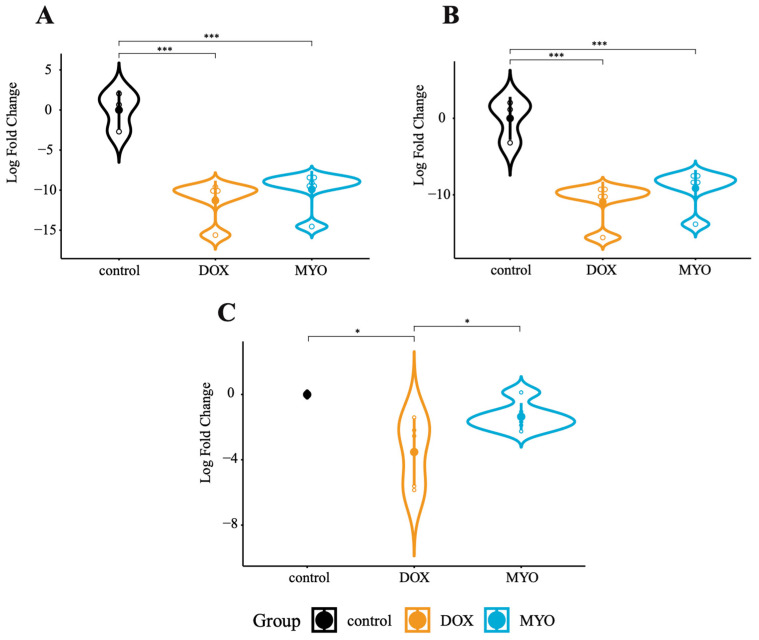
Validation of our RNA-seq data using qPCR (porcine heart tissue). (**A**) Differential expression of circ-MT:3033|3289 between control, DOX, and MYO animals. (**B**) Differential expression of circ-MT:3070|3478 between control, DOX, and MYO animals. (**C**) Differential expression of circ-16 between control, DOX, and MYO animals. * *p* < 0.05; *** *p* < 0.001. The error bars show standard deviations (SDs). Control: *n* = 3; DOX: *n* = 5; and MYO: *n* = 5. We performed differential expression analysis via one-way analysis of variance (ANOVA) of log2 fold changes, which were calculated with the ∆∆Ct method. For each combination (circRNA, tissue, and group), the data were checked for normality with the Shapiro–Wilk test. Furthermore, we tested for heteroscedasticity with Levene’s test. We performed a standard ANOVA (with pairwise *t*-tests for post hoc analysis) if its assumptions were met or the Kruskal–Wallis test (with a pairwise Wilcoxon signed-rank test for post hoc analysis) otherwise. The Benjamini–Hochberg method was used to correct for multiple testing.

**Figure 2 biomolecules-13-01711-f002:**
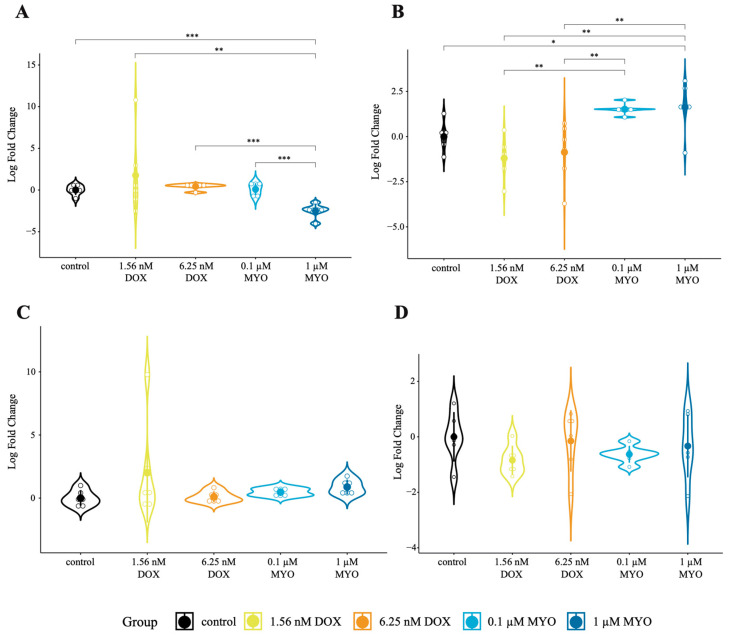
In vitro expression of mt-circRNAs in DOX- and MYO-treated cells. (**A**) Differential expression of circ-MT:3033|3289 in pCPCs between the control, DOX, and MYO groups. (**B**) Differential expression of circ-MT:3033|3289 in pCFs between the control, DOX, and MYO groups. (**C**) Differential expression of circ-MT:3070|3478 in pCPCs between the control, DOX, and MYO groups. (**D**) Differential expression of circ-MT:3070|3478 in pCFs between the control, DOX, and MYO groups. * *p* < 0.05, ** *p* < 0.01, and *** *p* < 0.001. The error bars show standard deviations (SDs). Each treatment group: *n* = 6. We performed differential expression analysis via one-way analysis of variance (ANOVA) of log2 fold changes, which were calculated with the ∆∆Ct method. For each combination (circRNA, cell type, and group), the data were checked for normality with the Shapiro–Wilk test. Furthermore, we tested for heteroscedasticity with Levene’s test. We performed a standard ANOVA (with pairwise *t*-tests for post hoc analysis) if its assumptions were met or the Kruskal–Wallis test (with a pairwise Wilcoxon signed-rank test for post hoc analysis) otherwise. The Benjamini–Hochberg method was used to correct for multiple testing.

**Figure 3 biomolecules-13-01711-f003:**
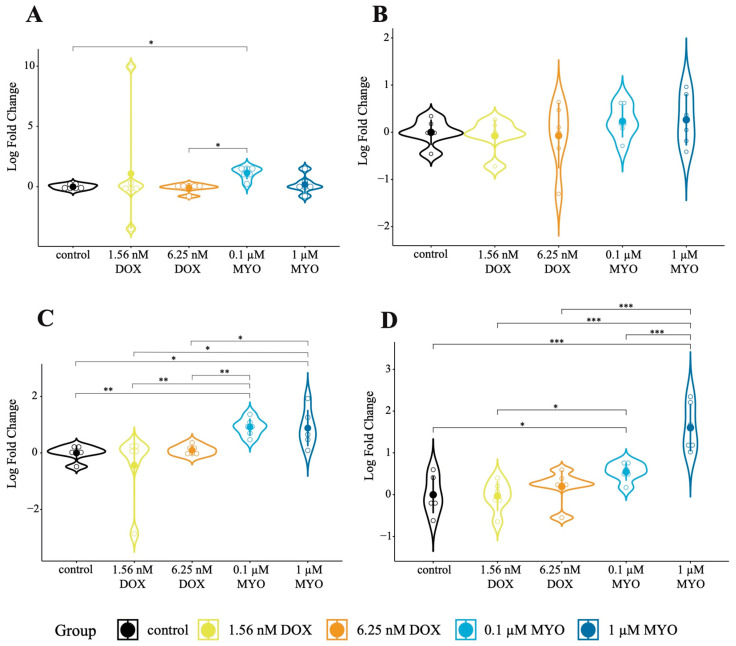
Cas3 and Ki67 expression in DOX- and MYO-treated cells in vitro. (**A**) Differential expression of Cas3 in pCPCs between the control, DOX, and MYO groups. (**B**) Differential expression of Cas3 in pCFs between the control, DOX, and MYO groups. (**C**) Differential expression of Ki67 in pCPCs between the control, DOX, and MYO groups. (**D**) Differential expression of Ki67 in pCFs between the control, DOX, and MYO groups. * *p* < 0.05, ** *p* < 0.01, and *** *p* < 0.001. The error bars show standard deviations (SDs). Each treatment group: *n* = 6. (pCPCs, porcine cardiac progenitor cells; pCFs, porcine cardiac fibroblasts.) We performed differential expression analysis via a one-way analysis of variance (ANOVA) of log2 fold changes, which were calculated with the ∆∆Ct method. For each combination (circRNA, cell type, and group), the data were checked for normality with the Shapiro–Wilk test. Furthermore, we tested for heteroscedasticity with Levene’s test. We performed a standard ANOVA (with pairwise *t*-tests for post hoc analysis) if its assumptions were met or the Kruskal–Wallis test (with a pairwise Wilcoxon signed-rank test for post hoc analysis) otherwise. The Benjamini–Hochberg method was used to correct for multiple testing.

**Figure 4 biomolecules-13-01711-f004:**
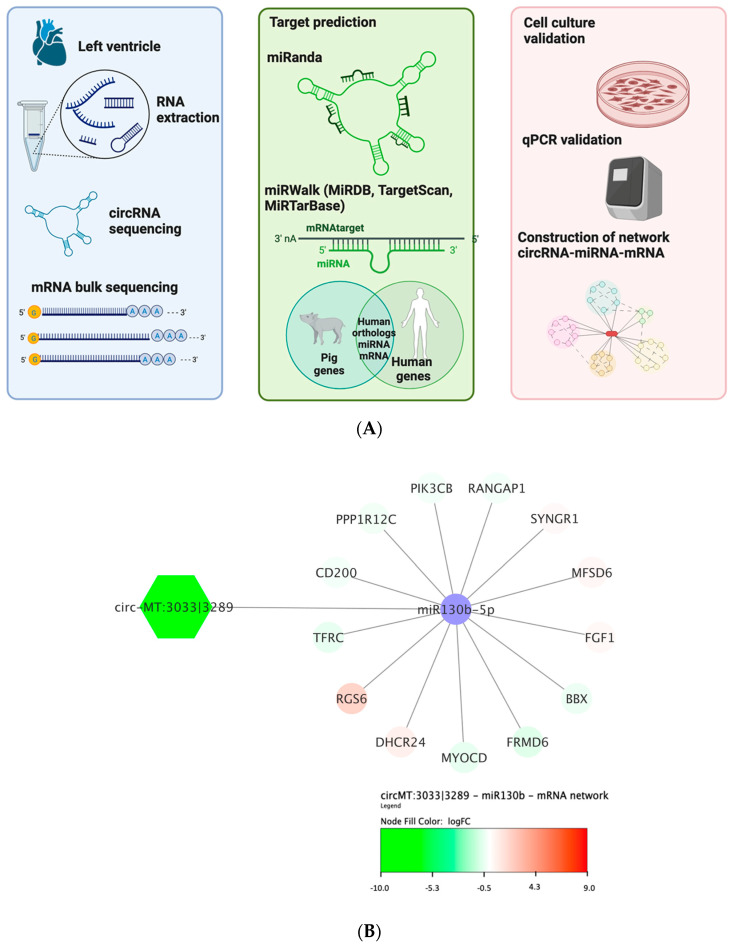
A network analysis and resulting network of mitochondrial circ-MT RNA/miRNA/mRNA interactions. (**A**) Graphical representation of the network analysis pipeline depicting the sequencing methods, target prediction algorithms, and validation techniques employed for constructing the circRNA–miRNA–mRNA network (created with BioRender.com). (**B**) Circ-MT RNA/miRNA/mRNA regulatory network constructed based on the MIRwalk-algorithm verified with bulk mRNA-seq data from the DOX and MYO animal models. The octagons represent significantly upregulated circRNAs in the MYO group compared with the DOX group with predicted miRNAs (the violet color) and mRNA targets validated using the bulk mRNA-seq approach. Gradual color changes indicate the log fold changes (LogFC) of significantly expressed genes (FDR < 0.05).

## Data Availability

The bulk mRNA-sequencing data discussed in this publication have been deposited in NCBI’s Gene Expression Omnibus and are accessible through GEO Series accession number GSE197049.

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
