# Peer review of "A CircRNA–miRNA–mRNA Network for Exploring Doxorubicin- and Myocet-Induced Cardiotoxicity in a Translational Porcine Model"

_biomolecules, 2023, doi:10.3390/biom13121711_

Round 1

Reviewer 1 Report

Comments and Suggestions for Authors

Dear Authors,

The presented manuscript embracing the identification of circRNA-miRNA-mRNA networks in induced cardiotoxicity in a pig model may be of a great value for further characterisation of underlying mechanisms and potential therapeutic targets. The text is nicely written and well organised. However, some important information is missing, and some parts are not clear enough. Detailed comments are below.

Line 133 – the description of the sequencing approaches is not clear, especially the naming. Total RNA can be used to sequence circRNAs and mRNAs too, while “bulk sequencing” does not necessarily mean only mRNA sequencing, so the used naming is not informative. Moreover, add the information about the sequencing depth of circRNAs, as you did for mRNAs in line 143. Please, specify if the sequencing depth given in lines 143-144 is for both end reads (7.5-10 million for one direction read) or for one direction reads?

Line 147 – in the described bioinformatics pipeline you mentioned circRNAs. Did you used the same pipeline and software for mRNAs identification? You mentioned bulk RNA-seq pipeline in line 154 but it is not clear enough. Moreover, provide information which annotation files were used exactly (line 153), because it is too vague. What is more, in lines 156-157, you indicated expression analysis between DOX and MYO groups, what about the control group? Where are the results of this differential expression analysis presented (the number of identified DEGs, FC, p-adj etc.)?

Lines 161-162 – the input for the Cytoscape analysis is not clear. What do you mean by “validated expression” and “fundamental input”? “The most significantly” stands for the most statistically significant (p-adj) or the highest fold change? How many such genes were chosen? Please, add appropriate information to make it more clear.

Lines 170-171 – the sentence is not grammatically correct so it is not clear.

Line 176 – “gener” - this word is not clear.

Line 190 – “RLT buffer” lacks the manufacturer.

Line 200 – How many and which genes/sequences were validated? On what basis they were chosen? How many technical repetitions of qPCR reactions were run?

Line 220 – please, add the information about the sequencing statistics for circRNA-seq and mRNA-seq (the obtained number of reads per sample or the obtained factual sequencing depths).

Line 232 – please, unify the naming of circRNAs because circRNA names and localization are different in the supplementary table S1, so it is not clear which circRNAs you investigated.

Subsections 3.3, 3.4 – in the lines 257-258 you stated that MYO treatment causes lower levels of cytotoxity than DOX, which was based on the observation that MYO could be administered in higher doses than DOX. And you stated in lines 20-21 that this caused by the liposomal encapsulation. In the subsections 3.3 and 3.4 you revealed that MYO treated cells had altered expression levels of some sequences when compared to DOX treated cells. These results led you to conclusions that “MYO treatment influences the expression of circ-MT:3033|3289 more significantly than DOX treatment” (lines 274-275) and “distinct effects of two chemotherapeuthic agents on apoptosis…” (lines 304-305).

I have some doubts regarding these results and drawn conclusions. Since MYO, thanks to the encapsulation, is less cytotoxic than DOX (as you mentioned), you could use higher MYO doses than DOX. Thus, the observed expression changes could be caused by the “dosage effect” and not the chemical nature of the used agent (as you mentioned in lines 20-21, MYO is the same chemical agent, but encapsulated). Thus, in my opinion, the obtained results do not clearly indicate “distinct effects” of MYO and DOX since they are based on the same chemical substance and were tested in different doses. Address this issue, please.

Subsection 3.5 – in this subsection, you described the identified network between circRNAs and miRNAs. In my opinion, this part is very superficial and does not support the drawn conclusions. First of all, the mentioned miRNAs were identified in silico and were not validated experimentally (miRNA-seq or qPCR), so conclusions highlighting “the prominent role” of a miRNA  in cardiotoxicity are too far-reaching (e.g. lines 343-344). Moreover, the interactions are described in a superficial manner without proposing possible explanation of the observed results, including the state of knowledge about the nature of such interactions. This may be exemplified by lines 327-331, where you stated that one upregulated circRNA and one downregulated circRNA had interactions with miRNAs which target detected downregulated mRNAs. What mRNAs exactly? Why specifically this interaction is so important? Upregulated circRNAs, acting as sponges, should cause miRNA expression level decrease and, as a result, the increase of target mRNA levels. This is only concordant with a part of your example, where one circRNA is down. But what is the mechanism behind the other part regarding the upregulated circRNA and how it can be connected to downregulated mRNAs via miRNAs? Thus, since miRNA levels were not investigated, it is not possible to state if they are truly engaged in the mentioned interactions in the indicated manner. It is also exemplified by the figure 4B, where upregulated circRNA was predicted to be associated with miR-130b-5p and its target genes. In line 356, you stated that the target genes are “validated using bulk RNA-seq approach”. However, this approach allowed you to validate their expression, not their nature as miRNA target genes, especially since some of these genes are upregulated and some are down. Furthermore, no direct interactions were confirmed between them and miRNAs using specifically designed experiments.

Line 339-340 – you stated that “these miRNAs target mt-circRNAs”. Which mt-circRNAs exactly?

Figure 4A – illegible.

Supplementary Figure S2 – I suggest adding some graphics along with nucleotide sequences, representing the identified back-spliced junctions to facilitate the interpretation of the chromatograms.

Line 376-377 – As mentioned before, the conclusion suggesting the identification of significantly regulated circRNA “together with relevant microRNAs and gene targets” is too far reaching since solely the expression levels of circRNAs and mRNAs were revealed.

Best wishes,

Reviewer 2 Report

Comments and Suggestions for Authors

The manuscript investigated circular RNA (circRNA) expression in pigs treated with either DOX, MYO, or control, and the authors developed a circRNA-miRNA-mRNA prediction network in the myocardium. They demonstrated the presence of circRNAs from RNA sequencing data in vivo and in vitro. The microRNAs, predicted to interact with circRNAs, were used for a mRNA prediction. The predicted mRNAs were intersected with mRNA data of the bulk RNA sequencing. The researchers linked the identified GO enrichment terms of the predicted mRNAs from the network to DNA-binding transcription activation and cell cycle regulation.

The study's main aim was to investigate the impact of DOX and MYO on circRNA expression and their role in cardiac damage; however, in the present form, they have shown a model for a potential regulation mechanism that needs further exploration. 

Major comments:

  1. The sequencing data of the 8 circRNAs that exhibited significant dysregulation between the DOX-control and MYO-control groups are in Supplementary Table S2 (lines 226-228, and 433). However, in the reviewer´s understanding, each circRNA should have two data rows based on the 2 groups that were statistically investigated, e.g., DOX vs. control and MYO vs. control (line 227). Additionally, the 3 circRNAs in Figure 1 are not within the 7 circRNAs in the Supplementary Table. Therefore, in the reviewer’s opinion, the RNA-seq data of those circRNAs from Figure 1 should also be included (this could justify their selection based on “most significant changing” circRNAs – line 224), and the total number of significant dysregulated RNA´s would then be 10 (7 circRNAs from Supplementary table S2 + 3 mt-circRNAs from Fig. 1). 
  2. Concerning the Supplementary Dataset S1, the authors state that they performed qPCR to measure the expression of 6 novel significantly regulated circRNAs (line 235) in myocardial tissue from each animal. However, it must be comprehensible that the circRNAs have been measured in different treatment groups in the first Excel sheet. Additionally, only 5 circRNAs have been measured. 
  3. Supplementary Dataset 4 shows the bulk RNA-seq dataset mRNAs intersecting with predicted miRNA targets. However, it needs to be comprehensible from which comparison the FC and p-values are derived. It would be recommended to show each gene's data from all comparisons (e.g., MYO vs. DOX, MYO vs. control, DOX vs. control). In line 319, the authors identified 126 overlapping genes from the predicted mRNAs and bulk RNA-seq data. However, in supplementary Dataset 4, the reviewer counts 121 genes without circRNAs and 128 with circRNAs. Also, it remains unknown in which groups these genes were detected (control/DOX/MYO) and in which comparison they were dysregulated. In the reviewer's opinion, both treatments should be investigated versus their control and then compared, e.g., counterregulatory mechanisms. 
  4. The author´s say that the upregulated circRNA1 and downregulated circRNA 4&6 sponge miR-17, miR-15b, and let7 isoform (lines 327-332). The precited mRNAs are predominantly downregulated; hence, the circRNAs reduce cardiotoxicity by this mechanism. However, since one of the circRNAs is upregulated, this conclusion may be too general and should be specified. 
  5. To the reviewer, the gene enrichment analysis in Supplementary Dataset 5 (sheet 2&3), is not performed with the validated mRNAs. It should be detailed which input was used (genes and number of genes etc.). 
  6. In the reviewer´s opinion, some experiments don´t align with the paper's aim, as measuring, e.g., Cas3 and Ki67 in cell cultures, doesn´t connect to exploring the regulatory effects of circRNAs. Also, the data does not seem to correlate with the cytotoxicity results, which should be included in the text. 
  7. The reviewer wondered whether, from the bulk RNA-seq data, miRNA expression could be determined and intersected with the predictions for the identified circRNAs. This could further validate the model. 
  8. The authors state in lines 443-444, that the prediction algorithm was validated in vivo and in vitro. In the reviewer´s opinion, the expression of some circRNAs have been validated in vivo (PCR confirmed NGS results), and 2 mt-circRNAs were validated in vitro. However, the predictions were validated in silico by intersecting with the bulk RNA-seq data.
  9. The authors previously published a paper treating pigs with DOX or MYO. The animals in their current study are derived from that study. However, the results of their prior study have not been discussed in connection with their current results. 

Minor comments: 

  1. In Section 3.1, the authors state that there is a significant difference in the expression of circ-7:22870230|22976632 in the MYO-control groups (p=0.048) (Fig. 1C) (line 233). However, this significance is not illustrated in Fig. 1C. Also, if there is a significant difference in MYO-control groups, excluding this circRNA from the following experiments is not comprehensible. 
  2. Section 3.3 should specify where the authors examined the expression of three circRNAs (line 263) because in the following paragraph and the corresponding Fig. 2, only two circRNAs were included. Furthermore, the author´s state that there is a statistical difference (p < 0.001) for MYO 0.1 μM compared to DOX 6.25 nM (circ-MT:3033|3289 in pCPCs). However, this is not illustrated in Fig. 2A. 
  3. For a better understanding of the workflow of circRNA selection for further analysis, including target predictions, the reviewer recommends making a graphical representation. 
  4. For the reviewer, it needs to be comprehensible why the backsplice junctions of the three mitochondrial circRNAs were mentioned at the end of the paragraph investigating two mt-circRNAs in cell culture models (lines 276-277). 
  5. Paragraph 3.4 failed to state that besides 1 uM MYO, 0.1 uM MYO was significantly increased compared to both Dox groups (lines 295-297). In the reviewer´s opinion, it is more important that this difference already appears at the lower dose. 
  6. In the reviewer´s opinion, the sentence in lines 303-305 does not represent the data shown in Fig. 3 because Cas3 expression did not change between DOX and MYO treatment in pCFs and only from 6.25nM DOX to 0.1 uM MYO in pCPCs. Therefore, the conclusion should not be generalized and limited to the presented data. 
  7. In line 320, the abbreviation DEG has been used for the first time. The reviewer recommends writing differential expressed genes, as this is the only time this abbreviation appears in the text. 
  8. In Line 343, the authors discuss Fig. 5B. However, they are presumably referring to Fig. 4B. Also, the wording of the sentence is misleading because 177 genes are not overlapping (based on the Supplementary material) with the bulk RNA-seq. This should be reformulated. 
  9. In general, the description in the supplementary material should be improved to specify, e.g., color-coding, gene input of GO, specific comparisons and statistics, etc...
  10. Supplementary dataset 3 has miRNA interactions for two mt-circRNA1. One is probably referring to mt-circRNA2. However, in Fig. 1, three mt-circRNAs were investigated, and it remains unclear why the third one is not included. It may be based on the PCR results, but this reasoning should be outlined. Also, it is not specified whether circ-MT:3033|3289 is mt-circRNA1 or 2 (same for the other mt-circRNAs).
  11. The authors disclosed that premature death occurred but seemingly did not affect the group size. They should detail how many animals have been enrolled and excluded in each group or if they have employed the animals with premature death for further analysis. 
  12. The authors selected the circRNAs based on the most significantly regulated expressions (lines 162-163). The reviewer recommends detailing the exact cut-off value to define “most significantly”.
  13. The reviewer understands that primary cells have been isolated, but the authors employed the term cell line (e.g., lines 181 and 384). Using the term primary cells is recommended as they differ from immortalized cell lines. 
  14. In line 322, the authors describe merging networks of up and downregulated circRNAs. However, the methodology of this merging is not explained in the Methods section. 
  15. The authors should specify which statistical test was used for which dataset, and it is recommended to include the statistical test in the figure legend. 
  16. The reviewer recommends to include the following references in the introduction: line 56-57 (PMID: 30934986, 33965565, 28460026).
  17. In line 61, “may sponge” is separated by more than one space. 
  18. In line 80, the authors say that MYO affected the expression of the two mt-circRNAs to a lesser extent than DOX. However, this is not statistically evident. The sentence from lines 78-80 is not comprehensible and should be reformulated. 
  19. In line 405, the authors say that mt-circRNA changes are only evident at specific doses. However, it may be essential to mention that this is true for a particular mt-circRNA.
  20. In line 425, “that” should be corrected to “than”.
  21. In line 425, the authors say that MYO influenced the expression to a greater extent than DOX. However, DOX didn´t affect the expression, so this phrase is misleading. 
  22. In Figure S1, the cytotoxicity of MYO treatment exceeds 100%. It needs to be explained to the reviewer what a cytotoxicity of more than 100% means. Also, the Methods do not explain how the data was normalized and a percentage was calculated. In the figure description, the n-number and employed statistical test is missing. Cell culture data should always be presented with standard error bars. Individual data points could add additional value. In the methods section, the authors disclosed the n-number but did not specify whether they were referring to technical – or biological repeats. Because they isolated primary cells, the number of successful isolations and the total number of animals used should be included.  
Comments on the Quality of English Language

Quality of english is acceptable

Reviewer 3 Report

Comments and Suggestions for Authors

Doxorubicin (DOX) is a widely employed chemotherapeutic agent for tumor treatment; however, it is associated with the potential for severe cardiac toxicity. The objective of this study is to elucidate the molecular underpinnings of doxorubicin-induced cardiotoxicity, focusing on circular RNA (circRNA) dynamics. Whole genome sequencing in a porcine model has been employed to address this objective. Nevertheless, several critical scientific issues necessitate further elucidation. 

1. Clarification of Data Quality and Analysis Process:

The research has yielded a relatively limited number of differentially expressed circRNAs through whole genome sequencing, notably fewer than typically obtained via RNA sequencing (RNA-seq). To enhance the comprehensibility of the findings, the authors must expound upon the data quality assurance measures adopted during genome sequencing and provide a meticulous account of their analytical procedures. 

2. Validation of Differential Expression circRNAs in Human Myocardial Cells:

Given the suboptimal homology of circRNAs across species, it is imperative for the study to bolster its clinical relevance by substantiating the observed differential circRNA expression patterns in the porcine model with evidence from human myocardial cells or cell lines. This validation is fundamental for extrapolating the findings to the clinical domain. 

3. Comparative Molecular Analysis of DOX and MYO:

To elucidate the mechanistic distinctions underpinning the differing cardiotoxicity profiles of DOX and MYO, it is recommended that the authors undertake a thorough molecular analysis and comparison of these compounds. Such an investigation may unveil the underlying factors contributing to the reduced cardiotoxicity associated with MYO, offering valuable insights for future therapeutic approaches.

Round 2

Reviewer 1 Report

Comments and Suggestions for Authors

Dear Authors,

Thank you for the extensive response to the previous comments. The manuscript has been improved. However, some issues were not addressed enough and some parts are still not clear. This especially applies to the description of methods and groups compared - it is not clear enough which parts were done in the previous study and the current one, as well as which comparisons were done previously, and, as a result, what was the main aim of this study. Detailed comments are in the pdf file.

Best wishes

Reviewer 3 Report

Comments and Suggestions for Authors

I have no other questions.

Author Response

We express our gratitude to the reviewer for their insightful comments, which have significantly contributed to the enhancement of our manuscript. We appreciate the time and effort the reviewer invested in the thorough revision of our work, culminating in the positive feedback received.

Best regards,

DL and JMT